# Comparison of Evolutionary Relationships between *Branchiostoma floridae*, *Ciona intestinalis*, and *Homo sapiens* Globins Provide Evidence of Gene Co-Option and Convergent Evolution

**DOI:** 10.3390/ijms242116009

**Published:** 2023-11-06

**Authors:** Nanako Yano, Toshifumi Minamoto, Hirosi Yamaguchi, Toshiyuki Goto, Takahito Nishikata

**Affiliations:** 1Faculty of Global Human Sciences, Kobe University, 3-11 Tsurukabuto, Nada-ku, Kobe 657-8501, Japan; 2151072h@cloud.kobe-u.jp (N.Y.); minamoto@people.kobe-u.ac.jp (T.M.); 2Graduate School of Human Development and Environment, Kobe University, 3-11 Tsurukabuto, Nada-ku, Kobe 657-8501, Japan; 3School of Science and Technology, Kwansei Gakuin University, 1 Gakuen Uegahara, Sanda 669-1337, Japan; hiroshi@kwansei.ac.jp; 4Frontiers of Innovative Research in Science and Technology (FIRST), Konan University, Minatojima-Minamimachi, Chuo-ku, Kobe 605-0047, Japan; toshiyuki.goto.2805@rabbit.kobe-u.ac.jp; 5RIKEN Center for Biosystems Dynamics Research, Minatojima-Minamimachi, Chuo-ku, Kobe 650-0047, Japan; 6Graduate School of Science, Technology and Innovation, Kobe University, Minatojima-Minamimachi, Chuo-ku, Kobe 650-0047, Japan; 7Research Institute for Human Health Science (RIH2S), Konan University, Minatojima-Minamimachi, Chuo-ku, Kobe 605-0047, Japan

**Keywords:** ascidian, hemoglobin, myoglobin, neuroglobin, cytoglobin, androglobin

## Abstract

Globins have been studied as model proteins to elucidate the principles of protein evolution. This was achieved by understanding the relationship between amino acid sequence, three-dimensional structure, physicochemical properties, and physiological function. Previous molecular phylogenies of chordate globin genes revealed the monophyletic evolution of urochordate globins and suggested convergent evolution. However, to provide evidence of convergent evolution, it is necessary to determine the physicochemical and functional similarities between vertebrates and urochordate globins. In this study, we determined the expression patterns of *Ciona* globin genes using real-time RT-PCR. Two genes (Gb-1 and Gb-2) were predominantly expressed in the branchial sac, heart, and hemocytes and were induced under hypoxia. Combined with the sequence analysis, our findings suggest that Gb-1/-2 correspond to vertebrate hemoglobin-α/-β. However, we did not find a robust similarity between Gb-3, Gb-4, and vertebrate globins. These results suggested that, even though *Ciona* globins obtained their unique functions differently from vertebrate globins, the two of them shared some physicochemical features and physiological functions. Our findings offer a good example for understanding the molecular mechanisms underlying gene co-option and convergence, which could lead to evolutionary innovations.

## 1. Introduction

Globins are small globular proteins containing approximately 150 amino acids. Their conserved feature is a 3-over-3 α-helical sandwich structure consisting of eight helices (named A through H) called the globin fold. The globin fold is also known as a heme prosthetic group (Fe^2+^-protoporphyrin IX)-binding site, which can reversibly bind oxygen and other ligands [1]. Moreover, globins form a large protein family present in all three domains of life, including bacteria, archaea, and eukaryotes [2]. Globins have long been studied as model proteins for elucidating the principles of evolution and establishment of protein architecture, which is achieved through the relationship between amino acid sequences, three-dimensional structures, physicochemical properties, and physiological functions [3].

In particular, vertebrate globins, including human hemoglobins, which play an essential role in oxygen transport and are related to diseases such as hemoglobinopathy and thalassemia, have a long history of research [4]. Recent comparative genomic studies revealed a diversity of globin genes in vertebrates, including hemoglobin, myoglobin, neuroglobin, cytoglobin, globin-E, globin-X, globin-Y, and androglobin, and their physiological functions and phylogenetic relationships [3]. Mammals, including humans, have five types of globins, namely hemoglobin, myoglobin, neuroglobin, cytoglobin, and androglobin. These five types are widely conserved in chordates, with a few exceptions, such as the loss of myoglobin in amphibians [5]. 

Vertebrate hemoglobins form hetero-tetramers composed of two α- and two β-type chains and are key players in oxygen transport via red blood cells [6]. The efficient oxygen transport by hemoglobin is due to the dynamic structural allostery of the tetramer [7]. Conversely, myoglobin, which resides in the cytoplasm of skeletal and cardiac muscle fibers as a monomer [8], stores oxygen and facilitates its diffusion into the mitochondria [9]. Neuroglobin and cytoglobin are relatively newly discovered globins. Neuroglobin is mainly expressed in the nervous system and has been suggested to play a protective role by sensing and controlling neuronal oxidative stress [10]. Cytoglobin is expressed in a wide variety of tissues, although it is predominantly expressed in fibroblasts and fibrotic cells, which are involved in collagen synthesis [11]. Androglobins are chimeric proteins that are approximately twice as large as other globins. They contain a globin fold structure and have oxygen-binding ability [12]. In terms of physicochemical properties, the co-ordination of Fe^2+^ within the heme prosthetic groups of hemoglobin and myoglobin is pentaco-ordinated, whereas those of neuroglobin and cytoglobin are hexaco-ordinated. These differences in co-ordination alter their binding affinities for oxygen and other ligands that are directly linked to their physiological functions [1]. However, the expression of all these globin genes, except androglobin, is induced under hypoxia, probably due to their functional roles in oxygen binding [11,12,13,14]. 

A wide variety of physiological roles and differential expression patterns of globin proteins are conserved within the chordate clade, suggesting that this was a result of adaptive genetic changes that occurred in the very early stages of vertebrate evolution. Moreover, phylogenetic analyses of the vertebrate hemoglobin-α and -β have revealed several examples of convergent evolution [15]. For example, the oxygen transport function evolved independently in jawed and jawless vertebrates, suggesting convergent evolution [16]. Thus, chordate globin genes offer good research materials for revealing how proteins and genes undergo adaptive changes during biological evolution.

Genome data of the cephalochordate (amphioxus, *Branchiostoma floridae*) and urochordate (tunicate, *Ciona intestinalis*) have been used to perform comparative phylogenetic studies on the evolution of globin families in the entire chordate clade [17,18]. Within the amphioxus genome, there are 15 globin genes consisting of the putative ancient forms of all five or eight vertebrate globin families [19]. The tunicate genome contains four globin genes that form a single clade [20]. These results confirm that, during chordate evolution, cephalochordates diverged first, with urochordates and vertebrates forming a sister group [21]. In addition, the independent evolution of urochordate globin genes has been suggested to be a convergent evolution [15]. Although sequence analyses have suggested the monophyletic evolution of urochordate globin genes, their physicochemical properties and physiological functions are not clear. Thus, in this study, we revisited *Ciona* genome data and determined the number of globin genes in the genome. The expression patterns of *Ciona* globin genes were revealed using real-time RT-PCR, and their physiological functions, including respiration and oxidative energy-producing functions, are discussed. By combining sequential and structural data with our functional data, the convergent evolution of *Ciona* globin genes was reconsidered, and we offer some evidence for the mechanism underlying protein adaptation.

## 2. Results

### 2.1. All Five Ciona Globins Were Monophyletic

To confirm the monophyletic evolution of *Ciona* globin genes, we re-examined the number of globin genes in the *Ciona* genome by searching NCBI and Ghost databases using the BLAST search program with human and ascidian globin amino acid sequences. As shown in Table 1, six different genes were identified and tentatively designated as Gb-1 to Gb-6. Gb-1 to Gb-5 were predicted to have a myoglobin-like domain (cd01040: Appendix A), which represented the 3-over-3 α-helical sandwich structure corresponding to the globin fold [22]. Moreover, they contain two conserved histidine residues, namely, the distal (helix E) and proximal (helix F) histidine residues, which co-ordinate a heme prosthetic group (Figure 1a). Their positions were similar to those of human cytoglobin and neuroglobin, but not those of hemoglobins and myoglobin (there were gaps in between helix E and F in our alignment), suggesting hexaco-ordinated heme proteins. Gb-1, -2, -3, and -5 encode proteins of similar sizes (approximately 150 aa), of which the myoglobin-like domain occupies almost all regions. Gb-4 encoded a longer protein (293 aa) with signal peptides (20 aa), followed by an uncharacterized sequence (133 aa) and a myoglobin-like domain (140 aa). Gb-6 is a chimeric gene that has a relatively long coding sequence and contains two conserved domains, the CysPc (cl00051: calpain protease domains IIa and IIb) and globin-like (cl21461) superfamily domains, and is suggested to be a homologous gene of chordate androglobins [23]. We could add one more globin gene (Gb-5) to previous studies and concluded that the *Ciona* genome contained five globin genes and one androglobin gene (Table 1). Thus, using the entire repertoire of globin proteins from *Ciona* genome, we re-examined the phylogenetic analysis. For phylogenetic analysis, we used the amino acid sequences of the following genes (Figure 1b): Gb-1 to -5 of *Ciona* globins, the 5 human globins representing all human globin families (hemoglobin-α, hemoglobin-β, myoglobin, cytoglobin, and neuroglobin), and the 12 amphioxus full-length globins reported by Ebner et al. (2010) [19]. Globin proteins of acorn worm, sea urchin, and sea stars were included to represent the hemichordates and echinoderms. Although we could find four other urochordate globin-like genes within the NCBI database, three of them were relatively long proteins that contained Adgb_C_mid-like domain (C-terminal middle region of androglobins) and the other was cytoglobin-2-like protein of *Styela clava*. Thus, we added this sequence as an example of urochordate globins since it was the only one except for *Ciona* globins. Plant globins were included as outgroups. This phylogenetic tree is consistent with previous studies on four *Ciona* globins (corresponding to our Gb-1 through -4: [15,20]). *Ciona* globins formed a single clade, even with Gb-5, and were the sister group of the human clade containing all human globins and hemichordate and echinoderm globins, although the bootstrap value of this node was relatively low. Thus, the diversification of *Ciona* globins could have occurred in parallel with that of human globins.

In our phylogenetic analysis, Gb-1 and Gb-2 were recently branched in the *Ciona* clade (Figure 1b). According to the genome database, these two are closely situated in the same scaffold, forming a gene cluster with only a 38 bp intergenic region (Table 1). Similarly, human hemoglobin-α and -β were recently branched in the human globin clade to form gene clusters. This similarity might suggest the convergent evolution between human hemoglobin-α/-β and *Ciona* Gb-1/-2.

### 2.2. Gb-1 and Gb-2 Expressed in Continuously Moving Tissues

To determine the functions of *Ciona* globins, we examined the expression profiles of five globin genes. Seven tissues, the brain (Br), branchial sac (BS), heart (Ht), hemocytes (Hc), body-wall muscle (Mu), ovary (Ov), and siphon (Sp), were surgically excised from healthy adults, and total RNAs were extracted and analyzed by real-time RT-PCR (Figure 2a). Gb-1 was primarily expressed in the branchial sac and heart. Gb-2 was primarily expressed in heart. Although it was not significant, Gb-2 showed a relatively high level of expression in branchial sac and hemocytes. The branchial sac contains many cilia that continuously wave to generate an inward flow of seawater. The heart also beats continuously, even though the pumping direction is flipped periodically because the ascidian has an open circulation system. Ascidian hemocytes are a heterogeneous population consisting of at least nine cell types, most of which can crawl [25]. Thus, Gb-1 and -2 are thought to be expressed mainly in tissues with continuous movement and high energy consumption. Interestingly, although the expression of Gb-1 was lower than that of Gb-2, the expression ratios of these two genes were approximately constant (Gb-2/Gb-1 in Br, BS, Hc, and Ht were 4.4, 3.1, 8.0, and 4.2, respectively). Since the difference between the expression of Gb-1 and Gb-2 may be due to the difference in the amplification efficiency of RT-PCR caused by the different primer sequences, the transcriptional control of these two genes may be the same. The fact that these two genes formed a cluster also supports this hypothesis. If these two translated products exist in similar amounts in the cell, they could form a hetero multimer, such as the tetramer of vertebrate hemoglobin-α and -β. Gb-3 and Gb-4 are expressed only in the siphon, which mainly consists of the muscle for opening and closing, sensory neurons for accepting physical or light stimulation, and the epidermis. However, their expression in the brain and body-wall muscles is extremely low. Thus, the tissue specificity of their expression remains uncertain.

### 2.3. Gb-1 and Gb-2 Were Induced by Hypoxia, While Gb-3 and Gb-4 Were Not

The induction of globin gene expression under hypoxic conditions is a key feature in understanding globin gene function. For example, the expression of human globin genes including hemoglobin-α and -β, myoglobin, cytoglobin, and neuroglobin, which are suggested to have oxygen-binding abilities, can be induced under hypoxia [11,13,14]. Thus, we placed *Ciona* adults in degassed seawater for two hours and excised the brain (Br-), branchial sac (BS-), heart (Ht-), hemocytes (Hc-), body-wall muscle (Mu-), ovary (Ov-), and siphons (Sp-). The expression of Gb-1 and Gb-2 was upregulated about 10-fold in all tissues under hypoxia. Moreover, the expression of Gb-1 and Gb-2 was increased to approximately the same level in all tissues (Gb-1:Gb-2 in Br, BS, Hc, and Ht were 19.2:16.6, 10.9:11.0, 26.5:20.7, and 9.5:7.7, respectively). Since the expression of each gene was evaluated using the same primer set, these quantifications were reliable. The similar induction levels of Gb-1 and Gb-2 in each tissue under hypoxia support the idea that the transcription of these two genes is similarly regulated. In contrast, the expression of Gb-3 and Gb-4 was not induced under hypoxia. 

Moreover, we examined the expression profiles during early development. The expression of all globin genes increased as the developmental stages progressed, although the expression in the gastrula stage was very low. The expression level of Gb-2 in swimming larvae was comparable to that in hypoxic conditions of the heart, suggesting that this expression level was the highest. Empirically, oxygen depletion causes developmental delays and abnormalities during the early developmental stages; therefore, oxygen consumption is thought to be high during the developmental process. In particular, from the tail bud stage to the swimming larval stage, tail muscle cells and the central and peripheral nervous systems develop and become active. High levels of globin gene expression may support the energy consumption of these cells by providing sufficient levels of oxygen.

Although Gb-5 showed low expression in unfertilized eggs and swimming larvae, its expression remained low from eggs to adults. Thus, Gb-5 is thought to have little impact on *Ciona*’s entire life cycle.

## 3. Discussion

In the present study, we identified at least six globin superfamily genes in the *Ciona* genome. Gb-6 had already shown to be an androglobin homologue with 1554 amino acid residues by the previous sequence analyses [18]. Chordate androglobin is a chimeric protein with calpain-homolog and globin-like domains of approximately 1500 aa. Androglobin is predominantly expressed in mammalian testes and has been implicated in ciliogenesis but not in oxygen consumption [12,23]. Although Gb-6 was omitted from our phylogenetic analysis, its function in *Ciona* is intriguing.

Here, we present several features of Gb-1 and Gb-2. First, their sequence homology was the highest (59% amino acid sequence identity) among the *Ciona* globin genes and they are located on the same chromosome very close to each other, suggesting that they are a result of a recent duplication. Second, the expression of both could be induced by hypoxia. Third, they could make a heteromultimer, since they were supposed to be under similar transcriptional control and, thus, their molecular ratio should be stable. These features are also common to human hemoglobin-α and -β [1,5,26,27], supporting the convergent evolution of *Ciona* Gb-1/-2 and human hemoglobin-α/-β. However, Gb-1 and Gb-2 were predominantly expressed in the branchial sac and heart, and their expression in the hemocytes was much lower. It is unclear whether Gb-1 and Gb-2, expressed in blood cells such as hemoglobins in vertebrates, are involved in oxygen transport through blood circulation. 

Vertebrate myoglobin and neuroglobin are predominantly expressed in cardiac and skeletal muscles and the nervous system, respectively, and are thought to be important for oxygen transport and storage within the cytoplasm, supporting high energy consumption [8,9,10]. However, in addition to the ubiquitous expression of cytoglobin [28], the expression of myoglobin and neuroglobin has been observed in various cell types, suggesting that they have various physiological functions [29,30,31]. Although some of these functions are not directly related to the oxygen-binding ability, their expression is induced by hypoxia [11,13,14]. In this study, although we could not predict the functions of Gb-3 or Gb-4, their insensitivity to hypoxia suggests that their functions may be different from those of human myoglobin, neuroglobin, and cytoglobin. This means that Gb-3 and Gb-4 perform unique functions that help in the adaptation of *Ciona*.

In contrast, Gb-4 has an N-terminal signal peptide sequence with a cleavage site (Appendix A), suggesting that it may be a secreted protein. Although the physiological functions of extracellular hemoglobin released by the hemolysis of red blood cells have been discussed [32], no secretion of globin protein, which has a signal peptide sequence, has been reported in vertebrates. On the other hand, in the vascular fluid, hemolymph, or coelomic fluid of some invertebrate phyla, including Annelida, Nematoda, Arthropoda, and Mollusca [33,34], extracellular hemoproteins of various forms and sizes such as monomeric hemoglobin and giant (ca. 3600 kDa) hexagonal bilayer hemoglobin have been reported [35]. Although the existence of Gb-4 in *Ciona* hemolymph or coelomic fluid could not be revealed in this study, we reconfirmed the prediction of N-terminal signal peptide and its cleavage site using SignalP-5.0 [36]. If Gb-4 is proven to be a secreted protein, this provides strong evidence of the convergent evolution of extracellular globins in invertebrates and urochordate.

Although the tertiary structure of the main chain folding of globin proteins is relatively robust against amino acid sequence changes during molecular evolution [37], globin proteins have acquired a wide variety of cellular functions adapted to lifestyles. The physicochemical properties and regulatory mechanisms are important for these variable physiological functions. The monophyletic evolution of *Ciona* globin genes indicates that nucleotide and amino acid sequence diversification occurred in parallel with the vertebrate clade. This study revealed that some ascidian globins shared their physicochemical properties and regulatory mechanisms with vertebrate globins, suggesting convergent evolution. In contrast, other ascidian globins acquired unique functions in the ascidian lifestyle, meaning simple gene co-option. However, many questions about the evolutionary mechanisms of proteins are yet to be answered. The ascidian genome has not undergone two successive whole-genome duplications during vertebrate evolution and has few paralogous genes. The six globin genes examined in this study (including androglobin) are thought to be a full repertoire of the *Ciona* globin superfamily genes. Revealing the physiological functions and physicochemical properties of all six globins and comparing them to other chordate globins could offer a good example for understanding the mechanisms underlying gene co-option and convergent evolution, which would give rise to evolutionary innovation [15]. 

## 4. Materials and Methods

### 4.1. Experimental Animal

Adult sea squirts (*Ciona intestinalis* Type A) were provided by the National Bio-Resource Project (MEXT, Tokyo, Japan). After collecting eggs from adult sea squirts, chorions were removed [38] and the eggs were developed in filtered seawater at 18 °C. Embryos developed into gastrula and middle tail buds at 5 h and 12 h, respectively, and hatched approximately 18 h after fertilization. The hypoxic condition of adult *Ciona* was induced by 2 h of incubation with hypoxic seawater, which was autoclaved and aspirated 2 h before use. 

### 4.2. Quest for the New Ciona GLOBIN Genes and Phylogenic Analyses

Four globin genes have been described in *Ciona intestinalis* genome by Ebner et al. (2003) [20]: *Ciona intestinalis* cytoglobin-1 (hb1; 100183004), *Ciona intestinalis* neuroglobin-like (hb2; 100183005), *Ciona intestinalis* globin (hb3; 445724), and *Ciona intestinalis* globin precursor (hb4; 445726). We searched the National Center for Biotechnology Information (NCBI) and Ghost databases [39] using the online Basic Local Alignment Search Tool (BLAST) algorithm program and found two other globin homologous genes, *Ciona intestinalis* androglobin (100181831) and *Ciona intestinalis* uncharacterized protein (100181975). Thus, we tentatively named these genes Gb-1 to Gb-6, as shown in Table 1. 

The following 30 deuterostome globin genes were used for phylogenic analyses: five *Ciona* globin genes, excluding the androglobin-like Gb-6, and one *Styela clava* cytoglobin-2-like (XP_039264784) as urochordate globins; five typical globins representing each of five human globin families, hemoglobin-α (HBA1; 3039), hemoglobin-β (HBB; 3043), myoglobin (MB; 4151), cytoglobin (CYGB; 114757), and neuroglobin (NGB; 58157); 12 amphioxus globins (118413968, 118413972, 118430442, 118403510, 118419696, 118412603, 118419699, 118412671, 118430393, 118405873, 118429879, and 118414084), for which the full amino acid sequences are available, were picked up from 15 sequences reported by Ebner et al. (2010) [19]. Acorn worm (*Saccoglossus kowalevskii*) neuroglobin-like (1; XP_006822138.1), cytoglobin-1-like (2; XP_006822139.1), globin-1-like (3; XP_006822018.1), and extracellular globin-E1-like (4; XP_002733371.1) representing hemichordate, sea urchin (*Strongylocentrotus purpuratus*) myoglobin-like protein (XP_030855013.1), and sea stars (*Patiria miniate* and *Asterias rubens*) cytoglobin-like proteins (XP_038065996.1 and XP_033630012.1, respectively) representing echinoderms were included. Plant (*Arabidopsis thaliana*) hemoglobin 1 and 2 (1; NP_179204.1 and 2; NP_187663.1, respectively) were included as outgroups. Multiple alignments of amino acid sequences of these globin genes were executed by MAFFT with G-INS-1 strategy (progressive method with an accurate guide tree) (https://mafft.cbrc.jp/alignment/server/ accessed on 17 October 2023 [40]) The phylogram was constructed using neighbor-joining method with 88 aa gap-free sites.

### 4.3. Primer Design

Several putative primer sets for each globin gene were designed using Primer3 [41] and Primer-BLAST [42] and synthesized (Fasmac, Atsugi, Japan). After confirming reliable and specific amplification by real-time RT-PCR using acrylamide gel electrophoresis, the most suitable primer set was determined. The primer sequences for Gb-1, Gb-2, Gb-3, Gb-4, Gb-5, and cytoplasmic actin-3 (CA3; XM_002129028) are as follows: 

Gb-1(Hb1-3);

Primer-F: 5′-CGAAAAGGTGTCCCATCATGC-3′

Primer-R: 5′-GGTCTGATGAGTTAGGCGCA-3′

Gb-2 (Hb2-1);

Primer-F: 5′-TTGCCACTTCCATTTCCACG-3′

Primer-R: 5′-GCTCCACATTATGACCAGCG-3′

Gb-3 (Hb3-2);

Primer-F: 5′- GATGTCCCCGAAACTCTTCCA-3′

Primer-R: 5′-TGCCTCACGACTTTGGCATT-3′

Gb-4 (Hb4-4);

Primer-F: 5′-TTTGACTACATGGGCCCTGT-3′ 

Primer-R: 5′-ATGCTGACAGAGTAGGCGAG -3′

Gb-5 (Hb5-8);

Primer-F: 5′-TGCGGTATTCAGTCATCTCG-3′

Primer-R: 5′-CCCTATACTGGCAAGGTCAGA-3′

CA3;

Primer-F: 5′-GAAAGGAGGGTTTCAGGAG-3′

Primer-R: 5′-GATCCTCCAGCAAGAACG-3′

### 4.4. RNA Extraction and cDNA Synthesis

Tissues were excised from 12 *Ciona* adults. They were mixed and weighed for RNA extraction. Total RNA was extracted using an RNeasy Plus mini kit (Qiagen, Tokyo, Japan) according to the manufacturer’s instructions, and cDNA synthesis was performed using Superscript IV reverse transcriptase (Thermo Fisher, Waltham, MA, USA). The concentrations of RNA and cDNAs were measured using a NanoDrop1000 Spectrophotometer (Thermo Fisher). The cDNA samples were stored at −70 °C until further use.

### 4.5. Real-Time RT-PCR

The mRNA expression levels of five ascidian globin genes were evaluated by RT-PCR using QuantStudio^®^ 3 (Thermo Fisher). Real-time PCR was carried out using PowerUp™ SYBR™ Green Master Mix (Thermo Fisher) following the manufacturer’s instructions with 100 ng cDNA. The amplification program included an initial denaturation at 95 °C for 10 min and 40 cycles of denaturation at 95 °C for 15 s and annealing and extension at 60 °C for 1 min. Three PCR replicates were performed with the same cDNA samples. Relative expression levels were calculated as follows:Relative expression level = 2^−ΔCt^
∆Ct = Ct (gene of interest) − Ct (inner control gene)

Cytoskeletal actin (CA3; *Ciona intestinalis* actin-3) was used as an internal control.

### 4.6. Statistical Analyses

Significant differences between groups were determined by one-way ANOVA and Tukey post hoc test.

## 5. Conclusions

We identified six globin genes in the *Ciona* genome, and our phylogenic analysis of Gb-1 through Gb-5 confirmed the previous studies [18,22]. In this study, we described the expression profile of Gb-1 through Gb-5, including their inducibility under hypoxic conditions, suggesting the similarities between Gb-1/-2 and vertebrate hemoglobin-α/-β in various aspects. Our data confirmed the monophyletic evolution of *Ciona* globins and provided some good examples of the physicochemically and physiologically convergent and co-operative evolution of globin genes and proteins.

## Figures and Tables

**Figure 1 ijms-24-16009-f001:**
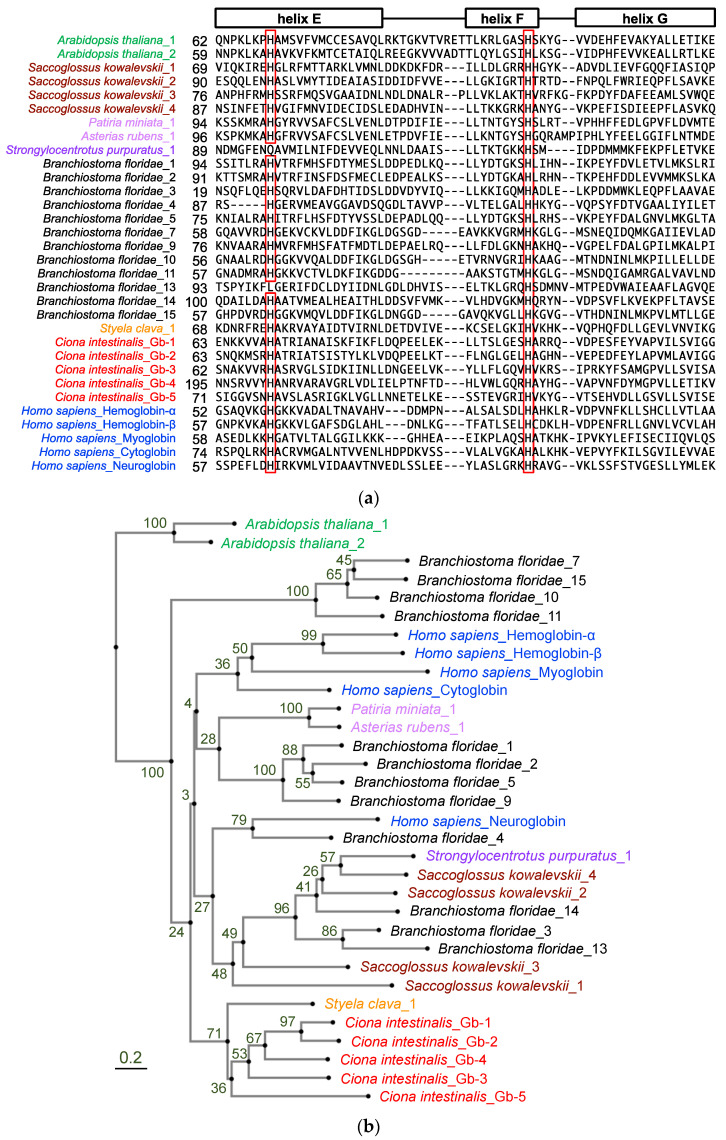
Phylogenic analysis of deuterostome globin genes. (**a**) Amino acid sequence alignment of human (blue: *Homo sapiens*), cephalochordate (black: *Branchiostoma japonicum*), urochordate (orange: *Styela clava*, red: *Ciona intestinalis* type A), echinoderm (light purple: *Patiria miniata* and *Asterias rubens*, purple: *Strongylocentrotus purpuratus*), hemichordate (dark red: *Saccoglossus kowalevskii*), and plant (green: *Arabidopsis thaliana*) globins. The leftmost amino acid number of each protein is indicated. This region represents helices E, F, and G of the globin fold. Highly conserved histidine residues, which are so-called distal (helix E) and proximal (helix F) histidine residues co-ordinating the heme prosthetic group, are represented in red rectangles. (**b**) The phylogram was constructed using neighbor-joining method with 88 aa gap-free sites and rooted with plant globins on the aLeaves web server (http://aleaves.cdb.riken.jp/ accessed on 17 October 2023 [24]). Values at nodes are bootstrap values from 100 replicates.

**Figure 2 ijms-24-16009-f002:**
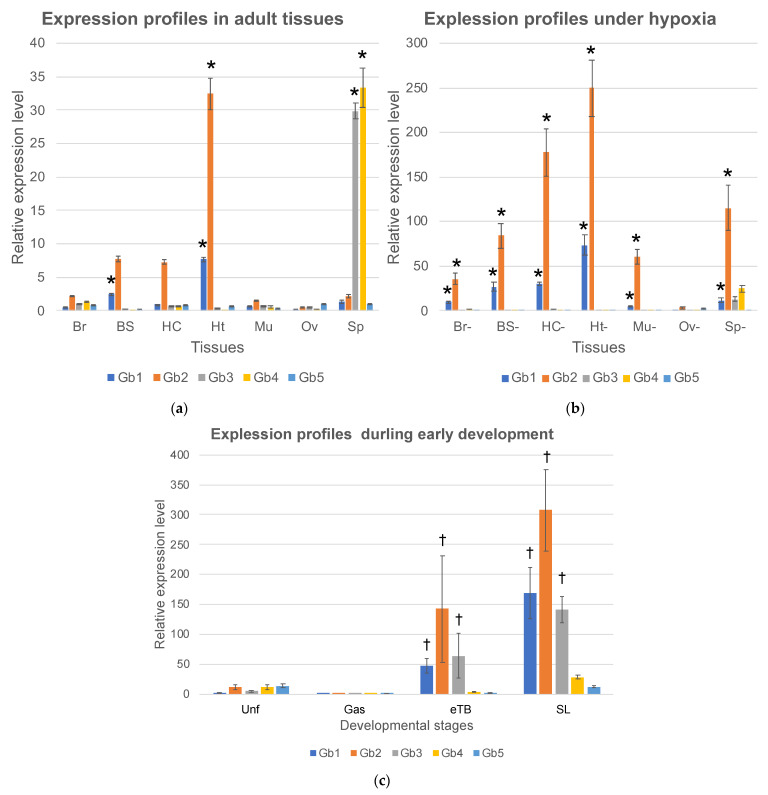
Expression levels of five *Ciona* globin genes (Hb 1–5) analyzed by real-time RT-PCR. The vertical axis represents the relative gene expression given by two to the power of (-ΔCt) compared to the cytoskeletal actin (CA3). Error bars represent SE. (**a**) Seven adult tissues, brain (Br), branchial sac (BS), heart (Ht), hemocyte (Hc), body-wall muscle (Mu), ovary (Ov), and siphon (Sp), were compared. *; a significant difference between tissues (*p* < 0.05) of each globin. (**b**) Seven adult tissues in hypoxic condition, brain (Br-), branchial sac (BS-), hemocyte (Hc-), heart (Ht-), ovary (Ov-), siphon (Sp-), and muscle (Mu-), were compared. *; a significant increase from normoxic to hypoxic conditions (*p* < 0.05) of each globin. (**c**) Unfertilized eggs (Unf) and embryos during early development, gastrula (Gas), early tailbud (eTB), and swimming larvae (SL) were compared. †; significant difference between the two groups, unfertilized egg/gastrula and tail-bud/swimming larvae (*p* < 0.05) of each globin. Note that the scales of vertical axes in a-c are largely different.

**Table 1 ijms-24-16009-t001:** Summary table of *Ciona* globin genes in databases.

This Study		NCBI				
Gene name	Suggested similarity	Gene ID	Gene symbol	Description	Location	Synonym
Gb-1	hemoglobin α/β	100183004	LOC100183004	cytoglobin-1	NW_004190356.2 (222268..229486)	hb1, CinHb1 *
Gb-2	hemoglobin α/β	100183005	LOC100183005	neuroglobin-like	NW_004190356.2 (220381..222229)	hb2, CinHb2 *
Gb-3	myoglobin/cytoglobin?	445724	hb3	globin	Ch. 3, NC_020168.2 (5923997..5925496)	hb3, CinHb3 *
Gb-4	invertebrate extracellular globin	445726	hb4	globin	Ch. 2, NC_020167.2 (2194984..2197811)	hb4, CinHb4 *
Gb-5	neuroglobin/pseudogene?	100181975	LOC100181975	uncharacterized	NW_004190472.2 (70166..73938)	
Gb-6	androglobin	100181831	LOC100181831	androglobin	Ch. 12, NC_020177.2 (4003794..4017836)	

*: Ebner et al., 2003 [20].

## Data Availability

The data presented in this study are available in Appendix A.

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
