# Peer review of "Comparison of Evolutionary Relationships between Branchiostoma floridae, Ciona intestinalis, and Homo sapiens Globins Provide Evidence of Gene Co-Option and Convergent Evolution"

_ijms, 2023, doi:10.3390/ijms242116009_

Round 1

Reviewer 1 Report

Comments and Suggestions for Authors

Line 277:  Please explain the basis for your statement that Gb-6 is an androglobin homolog. It is not in the tree so how can you say this? If this is based on a simple blast search, say so.

Line 277: “However, many questions about the evolutionary mechanisms of proteins are yet to be answered.” I do not grasp the meaning of this sentence.

Author Response

Thank you very much for your valuable comments. Our answers will appear in the following one by one. 

Line 277:  Please explain the basis for your statement that Gb-6 is an androglobin homolog. It is not in the tree so how can you say this? If this is based on a simple blast search, say so.

Ans: Thank you very much for pointing out of our insufficiency. The conclusions regarding Gb-6 were drawn solely from the information available in the database. Thus, we rewrite the discussion (L292-294).

Line 277: “However, many questions about the evolutionary mechanisms of proteins are yet to be answered.” I do not grasp the meaning of this sentence.

Ans: Thank you very much for pointing out of our insufficiency. We deleted this sentence.

Reviewer 2 Report

Comments and Suggestions for Authors

Dear Authors:

   The work presented by Yano et al. provides a comparative study of globin genes in a tunicate, Ciona intestinalis, by means of expression (RT-PCR) and phylogenetic analysis. This work, together with the study of the sequence of these genes, suggests that some of them have physiological and physicochemical functions shared with globins of vertebrates. The study is therefore of interest for publication in the journal.

   However, there are some points that need to be improved before being accepted for publication.

   In general, the work is not particularly profuse in analysis and results. The sequences are obtained from databases, from which a phylogenetic tree is made. On the other hand, a more in-depth RT-PCR study is carried out, in which globin expression is analysed in different tissues and oxygen conditions, as well as in different larval stages. This is reflected in  2 figures with results: an alignment as a basis for a phylogenetic tree with not many species, and relative expression levels graphs.

   Phylogenetic analysis: I consider that due to the small number of globins analysed, the conclusions that can be drawn are limited. The analysis (alignments and phylogeny) should be done with more globin sequences from a larger number of species. There are other tunicate species (Oikopleura dioica among others..), and of course vertebrates and invertebrates, which are fully sequenced and available in databases. Therefore, globins from more species of different taxonomic grades should be included, to give more robustness to the results obtained. This extended analysis would be bioinformatic, so it could be carried out without too much additional cost and would allow a much more robust analysis (Figure 1).

  2.- Expression analysis: in the graphs, the bars should show the mean and standard error (SE) derived from the experiments performed in triplicate. In addition, to compare results, statistical analyses such as one-way ANOVAs followed by Duncan's test or similar should be performed.

  3.- Figure 2 has three parts, whose letters a), b) and c) must appear in the figure itself.

    4.- Material and methods: The alignment algorithm used, ClustalW, has been very improvable for years. It is always advisable to use others that allow much more precise alignment, especially in the case of these globins, which are paralogous, and whose polymorphism is very high between species and between globins. Among the best alignment programmes is MAFFT or Muscle. The results could even be different and the phylogeny could also be different, as the phylogeny is based on the quality of the alignments performed.

   5.- Lines 360-361: if they have omitted data from other cDNA sets, perhaps this sentence could also be omitted.

   Therefore, the consideration of this manuscript in its current state would, in my opinion, require major revision.

Author Response

Thank you very much for your valuable comments. Our answers will appear in the following one by one. 

1.- Phylogenetic analysis: I consider that due to the small number of globins analysed, the conclusions that can be drawn are limited. The analysis (alignments and phylogeny) should be done with more globin sequences from a larger number of species. There are other tunicate species (Oikopleura dioica among others..), and of course vertebrates and invertebrates, which are fully sequenced and available in databases. Therefore, globins from more species of different taxonomic grades should be included, to give more robustness to the results obtained. This extended analysis would be bioinformatic, so it could be carried out without too much additional cost and would allow a much more robust analysis (Figure 1).

Ans: Thank you very much for your valuable suggestions. As you pointed out, our main concern with this manuscript is adding some evidence for the functional role of Ciona globins to reveal the mechanisms of protein evolution. In this concept, phylogenetic analysis was the confirmation of previous studies. Since we added Gb-5 (previously, it was recorded as “uncharacterized”) as one of Ciona globins, we should confirm the previous phylogenetic studies, especially for the monophyletic evolution of Ciona globin genes. However, your suggestion is also very meaningful. So, we added several deuterostome globins, which were recorded as “globin” or “globin-like” in NCBI database. Styela clava globin was the only tunicate globin other than Ciona globins as mentioned in the Result section (L135-139). We could not find Oikopleura globins in NCBI data without BLAST search. Several hemichordate and echinoderm globins were also added as examples of these animal groups.

2.- Expression analysis: in the graphs, the bars should show the mean and standard error (SE) derived from the experiments performed in triplicate. In addition, to compare results, statistical analyses such as one-way ANOVAs followed by Duncan's test or similar should be performed.

Ans: Thank you very much for your valuable suggestion. As we stated in the M&M section (L421-436), our results were obtained from a single biological experiment with triplicate RT-PCR experiments. According to your suggestion, we statistically analyzed the reliability of PCR experiment. Even with this analysis, we had to change some results (for example, tissues of Gb-1 and Gb-2 expression in normoxia). Thank you very much for your appropriate suggestion. Although we deleted the statement of the repeated experiments according to your suggestion (of course it is quite reasonable), we have done three independent experiments with consistent results. Unfortunately, different PCR reagents and machines were used in these experiments, and it was difficult to analyze them together. However, as the change of globin expression with normoxia and hypoxia in our manuscript was large and evident, we thought that it was worthwhile publication.

3.- Figure 2 has three parts, whose letters a), b) and c) must appear in the figure itself.

Ans: Thank you very much for pointing out of our insufficiency. We added them.

4.- Material and methods: The alignment algorithm used, ClustalW, has been very improvable for years. It is always advisable to use others that allow much more precise alignment, especially in the case of these globins, which are paralogous, and whose polymorphism is very high between species and between globins. Among the best alignment programmes is MAFFT or Muscle. The results could even be different and the phylogeny could also be different, as the phylogeny is based on the quality of the alignments performed.

Ans: Thank you very much for your valuable suggestion. We reanalyzed and re-aligned using deuterostome globins with MAFFT. As you predicted, the result was slightly different from our previous result. However, the monophyletic evolution of Ciona globins was confirmed even including with another tunicate, Styela clava, globin. It strengthened the idea of convergent evolution of globin in the urochordate clade.

5.- Lines 360-361: if they have omitted data from other cDNA sets, perhaps this sentence could also be omitted.

Ans: Thank you very much for your valuable suggestion. We deleted these sentences.

Round 2

Reviewer 2 Report

Comments and Suggestions for Authors

Dear authors:

The manuscript has been significantly improved, after the modifications and changes made, in a satisfactory manner.

However, there are some minor revisions that I believe could improve the article.

1.- The new manuscript's title only talks about the genus of some of the species studied, it would be convenient to put a sentence that includes all the species studied in the work. In case they only want to highlight 3 of them (Branchiostoma floridae, Ciona intestinalis and Homo sapiens), the full scientific name should appear in the title of the paper.

2.- There are several typographical errors in some new paragraphs added to the manuscript. They should be revised.

Otherwise, with the exception of these minor comments, I consider the article suitable for publication.

Comments on the Quality of English Language

- -

Author Response

Dear reviewer,

Thank you very much for your valuable comments. Our answers will appear in the following one by one. 

Comments and Suggestions for Authors

1.- The new manuscript's title only talks about the genus of some of the species studied, it would be convenient to put a sentence that includes all the species studied in the work. In case they only want to highlight 3 of them (Branchiostoma floridae, Ciona intestinalis and Homo sapiens), the full scientific name should appear in the title of the paper.

Ans: Thank you very much for your valuable suggestions. As you pointed out, in this manuscript, our main concern was to compare three species (Branchiostoma floridae, Ciona intestinalis, and Homo sapiens), in which all globin genes in each genome are thought to be described at this point. It is important for asserting the monophyletic evolution of globin genes. Thus, per your suggestion, the genus name and specific epithet were listed in the manuscript title to clarify the species.

2.- There are several typographical errors in some new paragraphs added to the manuscript. They should be revised.

Ans: Thank you very much for pointing out our careless mistakes. We failed to notice because we only checked the manuscript under turning on Track-Change. We are very sorry for such many mistakes. This time, we carefully checked the entire manuscript, and typographical errors were corrected. These corrected texts were highlighted in yellow.

Your Comments on the Quality of English Language

  • -none

Thank you again for your kind help.

Sincerely yours,

Takahito NISHIKATA